# Generalization of FedAvg Under Constrained Polyak-Łojasiewicz Type Conditions: A Single Hidden Layer Neural Network Analysis

## Abstract

In this work, we study the optimization and the generalization performance of the widely used FedAvg algorithm for solving Federated Learning (FL) problems. We analyze the generalization performance of FedAvg by handling the optimization error and the Rademacher complexity. Towards handling optimization error, we propose novel constrained Polyak-Łojasiewicz (PL)-type conditions on the objective function that ensure existence of a global optimal to which FedAvg converges linearly after $\mathcal{O}(\log(1/\epsilon))$ rounds of communication, where $\epsilon$ is the desired optimality gap. Importantly, we demonstrate that a class of single hidden layer neural networks satisfies the proposed constrained PL-type conditions required to establish the linear convergence of FedAvg as long as $m > nK/d$, where $m$ is the width of the neural network, $K$ is the number of clients, $n$ is the number of samples at each client, and $d$ is the feature dimension. We then bound the Rademacher complexity for this class of neural networks and establish that both Rademacher complexity and the generalization error of FedAvg decrease at an optimal rate of $\mathcal{O}(1/\sqrt{n})$. We further show that increasing the number of clients $K$ decreases the generalization error at the rate of $\mathcal{O}(1/\sqrt{n} + 1/\sqrt{nK})$.

## 1 Introduction

Federated learning (FL) is a distributed learning paradigm where multiple client devices collaborate with the help of a server to solve a joint problem while keeping the data of each client private (Kairouz et al., 2021). A typical FL problem aims to solve $\min_{\boldsymbol{w}} \sum_{k=1}^{K} \Phi_k(\boldsymbol{w})$, where $\Phi_k(\boldsymbol{w})$ is the loss at the $k^{\text{th}}$ client and $\boldsymbol{w}$ refers to the joint model the clients aim to learn. A standard and most widely adopted algorithm to solve the FL problem is the Federated Averaging (FedAvg) algorithm first proposed in (McMahan et al., 2017). Consequently, the study of the convergence performance of FedAvg has received wide attention (Konečnỳ et al., 2015; Stich, 2018; McMahan et al., 2017; Li et al., 2020; Zhou & Cong, 2017b). However, when it comes to ensuring generalization guarantees for FedAvg, the problem has not received significant attention, partially because of the challenging nature of the problem (Mohri et al., 2019; Sun et al., 2023; Hu et al., 2022). To prove the generalization guarantees for FedAvg, we need to bound $(a)$ the **optimization error** (on empirical loss) achieved by FedAvg, and $(b)$ the **complexity measure** such as the Rademacher complexity of the model (Arora et al., 2019; Mohri et al., 2019; 2018). The major challenge in guaranteeing good generalization performance is to bound both $(a)$ and $(b)$ above, which are often contradictory, i.e., proving optimization guarantees usually rely on restrictive assumptions on the loss landscape like (strong)-convexity or Polyak-Łojasiewicz (PL) inequality to be satisfied over the entire parameter space Haddadpour et al. (2019); Haddadpour & Mahdavi (2019) while the Rademacher complexity is large for an unbounded parameter space [see Theorem 5.10 (Mohri et al., 2018)]. Therefore, bounding both $(a)$ and $(b)$ simultaneously is challenging, thereby making it difficult to provide satisfactory generalization guarantees for FedAvg. To address these challenges in this work:

➢ We first analyze the convergence of FedAvg and establish **linear convergence** under a **new set of assumptions** that are only required to be satisfied **locally**. Importantly, to highlight the practicality of the assumptions, we establish that the proposed **assumptions are naturally satisfied by a single hidden-layer Neural Network (NN)**.

➢ We then study the generalization guarantees of FedAvg for the single hidden-layer NN and show

that the proposed local assumptions lead to a **Rademacher complexity that goes down with the number of samples** $n$ **as** $\mathcal{O}(1/\sqrt{n})$. Specifically, our analysis captures the effects of local samples, the number of clients, and model sizes on the performance of the FedAvg algorithm.

In the following, we discuss specific challenges and the drawbacks of the current state-of-the-art with respect to challenges $(a)$ and $(b)$ discussed above.

**Convergence of FedAvg.** As discussed earlier, several works have analyzed the convergence performance of FedAvg under various settings. In the non-convex regime, multiple works have established the convergence of FedAvg to a stationary point (local optimal) (Konečný et al., 2015; Stich, 2018; McMahan et al., 2017; Li et al., 2020; Zhou & Cong, 2017b). However, the local optimal does not guarantee a small empirical loss, and hence cannot be used to provide generalization guarantees. Some works have shown convergence of FedAvg to global optimal but under restrictive assumptions of (strong) convexity (Stich, 2018; Qu et al., 2020). In Haddadpour et al. (2019), the authors provide convergence of FedAvg to the global optimal by imposing the PL condition on the objective function, which is unfortunately not satisfied by several loss functions (e.g., log-logistic loss) over the whole parameter space. Importantly, assuming that the PL inequality is satisfied globally (without any restriction on the parameter space Haddadpour & Mahdavi (2019)) leads to a large Rademacher complexity, thus leading to worse generalization guarantees. This leads to the following question:

> Q1: *Can we develop conditions that are satisfied locally (on a restricted parameter space) rather than globally and provide convergence guarantees for FedAvg? Are there models that satisfy such a condition?*

To address Q1, we provide *new weaker* conditions (a constrained variant of the PL-inequality) on the global and local loss functions. Importantly, we prove that there exists a globally optimal point within a ball of radius $\rho$ around initialization to which FedAvg converges linearly. Moreover, we also establish that there exist NN architectures that satisfy the conditions proposed in our work.

**Generalization guarantees for FedAvg:** The generalization performance of centralized machine learning algorithms has been extensively studied (Mohri et al., 2018; Bousquet & Elisseeff, 2002; Emami et al., 2020). However, the study of generalization guarantees of FL algorithms is rather limited (Mohri et al., 2019; Hu et al., 2022; Yuan et al., 2021a). Notably, these studies often overlook the impact of the optimization algorithm Sun et al. (2023), and often rely on assumptions like Binary loss Hu et al. (2022); Mohri et al. (2019) and the Bernstein condition (Yuan et al., 2021a). Additionally, generalization bounds for meta-learning and FL are established in Fallah et al. (2021); Chen et al. (2021) under stringent assumptions such as strong convexity and bounded loss functions. Recently, Sun et al. (2023) has investigated the generalization of FedAvg via the lens of uniform stability. We note that these analyses impose strong assumptions such as bounded gradient and heterogeneity on the data, which are usually not satisfied by many problems of practical interest. Moreover, the optimization guarantees provided in Sun et al. (2023) are weaker compared to the linear convergence established in our work. Based on the above observations, we ask the following main question:

> Q2 : *Can we provide generalization guarantees for FedAvg? If so, what is the impact of $(a)$ the number of samples per client, $(b)$ the model size, and $(c)$ the number of clients on the generalization performance?*

We address Q2 by deriving Rademacher complexity when each client employs a single hidden-layer NN for FedAvg implementation. We show that the local assumptions developed to address Q1 play an important role in bounding the Rademacher complexity for FedAvg. Importantly, our analysis captures the effect of data samples and NN size, and the number of clients on the generalization performance of FedAvg. It is worth mentioning that to address both Q1 and Q2, we *do not* make some standard assumptions that are typically used in many existing works Li et al. (2019); Stich (2018); Yu et al. (2019); Haddadpour et al. (2019); Qu et al. (2020); Woodworth et al. (2020a;b); Hu et al. (2022); Mohri et al. (2019) such as: $(i)$ (strongly) convex loss, $(ii)$ bounded loss, $(iii)$ bounded gradients $(iv)$ bounded heterogeneity, and $(v)$ interpolation [1]. In this work, we have not assumed the existence of a global optimal point; rather, it is part of our conclusion.

---

[1]Interpolation refers to the existence of a $\boldsymbol{w}^*$ such that $\Phi_{k,i}(\boldsymbol{w}^*) = 0$ for all $k \in [K]$ and $i \in [n]$.

**Contributions.** The major contributions of our work include:

➢ `Answer to Q1`: For the first time, we show that FedAvg converges linearly to the optimal solution (see Corollary 3.2) if the local loss functions at each client and the global loss function satisfy a novel local PL-type assumption introduced in Assumption 2.4. It is important to note that the existence of a global optimal in our analysis is a part of our conclusion, *not* an assumption. To the best of our knowledge, both conditions introduced in Assumption 2.4 are new. It is also worth noting that these conditions do not follow from any of the existing results, even in the special case of centralized setting, i.e., for $K = 1$ (Chatterjee, 2022; An & Lu, 2023). In addition, we also establish that a single hidden-layer NN satisfies the two conditions proposed in Assumption 2.4. Specifically, we establish the conditions on the width of the NN as a function of the number of samples, number of clients, and the feature dimension, and on the eigenvalues of the Jacobian of the loss functions (or the scaling factor of the final output layer) such that the proposed conditions are satisfied. To our knowledge, these results are novel (see Theorems 4.5).

➢ `Answer to Q2`: To address `Q2`, we derive an upper bound on the Rademacher complexity for a class of single hidden layer NNs by utilizing the fact that the FedAvg iterates stay within a $\rho$-ball around the initialization. We point out that this is made possible by the conditions provided in Assumption 2.4. In particular, we show that the Rademacher complexity approaches zero if the radius $\rho = \mathcal{O}(\sqrt{n})^2$ and $m = \mathcal{O}(n^3)$, where $n$ is the number of samples at each client and $m$ is the width of the NN. We show that the generalization error regardless of the data heterogeneity diminishes as $\mathcal{O}(1/\sqrt{n})$. We finally corroborate our theoretical findings through numerical experiments.

## 2 FEDAVG: ALGORITHM AND ASSUMPTIONS

As discussed in Section 1, FL aims to solve the following optimization problem:

$$\min_{\boldsymbol{w}} \left\{ \Phi(\boldsymbol{w}) := \frac{1}{K} \sum_{k=1}^{K} \Phi_k(\boldsymbol{w}) \right\}, \tag{1}$$

where $\Phi_k(\boldsymbol{w}) := \mathbb{E}_{(\boldsymbol{x},y)\sim\mathcal{D}_k} l_k(f_{\boldsymbol{w}}(\boldsymbol{x}), y)$ is the loss function at client $k \in [K]$. Here, $y \in \mathcal{Y}$ is the true label, and $f_{\boldsymbol{w}}(\boldsymbol{x})$ is the output of model $\boldsymbol{w} \in \mathbb{R}^{d'}$ for an input feature $\boldsymbol{x} \in \mathbb{R}^d$, and $l_k : \mathcal{Y} \times \mathcal{Y} \to \mathbb{R}^+$ is the loss function at the client $k \in [K]$. In the above, $d'$ is the dimension of the parameter space. The following algorithm captures the main steps of FedAvg (McMahan et al., 2017). In Algorithm 1, $\Phi_{k,i}(\boldsymbol{w}_k^{r,t})$ denotes the empirical loss function at client $k \in [N]$ computed using sample $i \in [n]$.

In this and the subsequent section, we answer `Q1` posed in Sec. 1. In particular, we provide a general condition for the above algorithm to converge to a global optimum and for the model parameters to stay within a closed ball of radius $\rho$. In the later sections, we show that this condition is, in fact, satisfied for a single hidden layer NN. Specifically, this constraint imposes a natural regularization of the NN which provides better generalization, as discussed later. To prove our claim, we make the following standard assumptions on the loss function Ji & Telgarsky (2018).

**Assumption 2.1** (L- Smoothness). *The loss functions $\Phi_k$ and $\Phi$ are assumed to be $L_k$-smooth and $L$-smooth, respectively, i.e., $\|\nabla\Phi_k(\boldsymbol{u}) - \nabla\Phi_k(\boldsymbol{v})\| \le L_k\|\boldsymbol{u} - \boldsymbol{v}\|$ for all $k \in [K]$ and $\|\nabla\Phi(\boldsymbol{u}) - \nabla\Phi(\boldsymbol{v})\| \le L\|\boldsymbol{u} - \boldsymbol{v}\|$ for all $\boldsymbol{u}$ and $\boldsymbol{v}$.*

**Assumption 2.2** (Samplewise Smoothness). *The loss functions $\Phi_{k,i}(\boldsymbol{w})$ are assumed to be $l_{k,i}$-sample-wise smooth, i.e., $\|\nabla\Phi_{k,i}(\boldsymbol{v})\|^2 \le 2l_{k,i}\Phi_{k,i}(\boldsymbol{v})$ for all $k \in [K]$ and $i \in [n]$.*

To define the major assumptions required for the convergence of FedAvg Algorithm 1, we need the following definition (Chatterjee, 2022).

**Definition 2.3.** *Let $f : \mathbb{R}^d \to \mathbb{R}^+$ be continuously differentiable function on closed ball $\mathbb{B}[\underline{\boldsymbol{w}}^0, \rho]$ with center at initialization $\underline{\boldsymbol{w}}^0 \in \mathbb{R}^d$ and radius $\rho > 0$. Define*

$$\alpha(\underline{\boldsymbol{w}}^0, \rho) := \inf_{\boldsymbol{w}\in\mathbb{B}[\underline{\boldsymbol{w}}^0,\rho]} \frac{\|\nabla f(\boldsymbol{w})\|^2}{f(\boldsymbol{w})} > 0. \tag{2}$$

Next, we state an important assumption that leads to linear convergence within a ball around initialization.

---

[2]This is the radius over which our new condition should be satisfied.

---

**Algorithm 1** FedAvg McMahan et al. (2017)

---

1: **Initialize**: $\{\boldsymbol{w}_k^{0,0} = \underline{\boldsymbol{w}}^0\}$, $\boldsymbol{w}_k \in \mathbb{R}^d$ for $k = 1, 2, \ldots, K$
2: **for** $r = 0, 1, \ldots, R - 1$ **do**
3:      Broadcast $\underline{\boldsymbol{w}}^r$ to all the clients $k \in [K]$
4:      **for** $\tau = 0, 1, \ldots, T - 1$ **do**
5:          **for** each client $k \in [K]$ **do**
6:              Sample a batch $\mathcal{B}_k^{r,t}$ of size $|\mathcal{B}_k^{r,t}| = b$
                **SGD** step on $\boldsymbol{w}_k^{r,t}$ for $k \in [K]$:
                $\boldsymbol{w}_k^{r,t+1} = \boldsymbol{w}_k^{r,t} - \eta \widehat{\nabla \Phi}_k(\boldsymbol{w}_k^{r,t})$
                // $\widehat{\nabla \Phi}_k(\boldsymbol{w}_k^{r,t}) := \frac{1}{b} \sum_{i \in \mathcal{B}^{r,t}} \nabla \Phi_{k,i}(\boldsymbol{w}_k^{r,t})$
7:          **end for**
8:      **end for**
9:      **Receive** $\boldsymbol{w}_k^{r,T}$ from nodes $k \in [K]$
10:     **Aggregation** step : $\underline{\boldsymbol{w}}^{r+1} = \frac{1}{K} \sum_{k \in [K]} \boldsymbol{w}_k^{r,T}$
11: **end for**

---

**Assumption 2.4.** *For some initialization $\underline{\boldsymbol{w}}^0$ and radius $\rho > 0$, we make the following assumptions on the local and global loss functions:*

     *1. The loss function at each client is assumed to satisfy (see Theorem E.1)*

$$32\Phi_k(\underline{\boldsymbol{w}}^0) < \rho^2 \alpha_k(\underline{\boldsymbol{w}}^0, \rho). \tag{3}$$

     *Here, $\alpha_k(\underline{\boldsymbol{w}}^0, \rho)$ is as defined in equation 2 but with $f(\cdot)$ replaced by $\Phi_k(\cdot)$.*

     *2. The global loss function is assumed to satisfy the following condition*

$$\sqrt{128 e l'_{\max} K \Phi(\underline{\boldsymbol{w}}^0)} < (1 - \zeta_\rho)\rho \alpha_g(\underline{\boldsymbol{w}}^0, \rho), \tag{4}$$

     *for some $\zeta_\rho \in (0, 1)$. Here, $\alpha_g(\underline{\boldsymbol{w}}^0, \rho)$ is as defined in equation 2 but with $f(\cdot)$ replaced by $\Phi(\cdot)$.*

**Remark 1.** *In general, two very critical assumptions are made in the literature while proving linear convergence: $(i)$ interpolation, i.e., there exists $\boldsymbol{w}^*$ such that $\Phi_i(\boldsymbol{w}^*) = 0$ for all samples $i \in [n]$ Liu et al. (2022); Li et al. (2019), and $(ii)$ strongly convex loss Li et al. (2019); Karimireddy et al. (2020) or loss function satisfying the PL-inequality Fan et al. (2023). Later, a relaxed version of PL-inequality called local PL or $PL^*$-inequality was proposed where the PL-inequality needs to be satisfied over a small ball around the initialization (see Liu et al. (2022); Oymak & Soltanolkotabi (2019). Despite this relaxation, it makes a critical assumption on the existence of the optimal $\boldsymbol{w}^*$ such that the loss $\Phi_i(\boldsymbol{w}^*) = 0$ for all samples $i \in [n]$-the interpolation regime. In our work, we argue that this assumption can be relaxed with our novel condition shown in Assumption 2.4. It is important to note that our condition is fundamentally different from the $PL^*$-inequality in the following way:*

- There is a stark difference between our proposed condition and the the PL-condition (or $PL^*$ condition), which is defined as $\|\nabla \Phi(\boldsymbol{w})\|^2 \geq \mu(\Phi(\boldsymbol{w}) - \Phi(\boldsymbol{w}^*))$ for all $\boldsymbol{w} \in \mathbb{R}^d$ (and $\boldsymbol{w} \in \mathbb{B}[\underline{\boldsymbol{w}}^0, \rho]$ for $PL^*$ condition). In the PL-condition (and local PL), the constants do not depend on the initialization and radius as the condition is universally satisfied. Another important assumption made in the local/global PL-condition is the existence of a global optimal point $\boldsymbol{w}^*$. In contrast, our proposed condition does not require this assumption; instead, we prove the existence of a global optimal point under our novel condition.

- It is important to note that the PL-condition must be satisfied over the entire parameter space, which can restrict its applicability to certain loss functions such as logistic loss Karimi et al. (2016). On the other hand, our novel condition is assumed only over a small neighborhood around the initialization, making it more broadly applicable. Later we show that parameters such as initialization and the radius $\rho$ can be chosen so that the condition is easily (compared to the PL inequality) satisfied.

In this work, we have shown that the proposed condition is satisfied for at least a single hidden layer neural network. In Chatterjee (2022), the authors have shown that the wide neural network satisfies

the constrained PL inequality for a single client setting. Therefore, we strongly believe that the proposed condition in our work will also be satisfied for wide neural networks.

## 3 CONVERGENCE ANALYSIS

In this section, we establish that the FedAvg Algorithm 1 achieves linear convergence to a global optimum under the set of assumptions introduced in Sec. 2. Importantly, note that the existence of this global optimum is established as a conclusion rather than an assumption. Moreover, unlike other works, we do not explicitly assume interpolation to establish linear convergence of FedAvg (Haddadpour et al., 2019; Stich, 2018). In particular, we establish a proof that the sufficient conditions stated in equation 2.4 not only guarantee the linear convergence of Algorithm 1 but also ensure the existence of an optimal point denoted as $\boldsymbol{w}^*$ within the closed ball $\mathbb{B}[\underline{\boldsymbol{w}}^0, \rho]$. The following theorem is a precise statement whose proof can be found in Appendix 3.1.

---

**Theorem 3.1.** *Assuming that there exists an initialization $\underline{\boldsymbol{w}}^0 \in \mathbb{R}^d$, and a radius $\rho > 0$ such that Assumptions 2.1 and 2.4 are satisfied by loss functions $\bar{\Phi}$ and $\Phi_k$ for $k \in [K]$, then FedAvg ensures that there exists a $\boldsymbol{w}^* \in \mathbb{B}[\underline{\boldsymbol{w}}^0, \rho]$ such that $\lim_{R \to \infty} \Phi(\underline{\boldsymbol{w}}^R) = \Phi(\boldsymbol{w}^*) = 0$ provided the learning rate*

$$\eta \leq \min\left\{ \frac{2}{\alpha_{\min}}, \frac{\alpha_{\min}}{4 L_{\max} l'_{\max}}, \frac{\alpha_{\min}}{2 L_{\max} l'_{\max}}, \frac{1}{T\sqrt{\Psi_0}}, \frac{8}{\alpha_g T}, \frac{\zeta_\rho \rho}{T\sqrt{\Psi_0}}, \Psi_1, \Psi_2 \right\},$$

*where $l'_{max} := \max_k l'_k := \max_i l_{k,i}$; $L_{\max} := \max_k L_k$ ; $\alpha_{min} := \min_{k \in [K]} \alpha_k$ ; $\Psi_0 := 2e l'_{\max} K \Phi(\underline{\boldsymbol{w}}^0)$ ; $\Psi_1 := \sqrt{\frac{3}{L_{\max} l'_{max}}}$ and $\Psi_2 := \min\left\{ \frac{\alpha_g \alpha_{\min}}{4T(4 L_{\max}^2 l'_{\max} + L l'_{\max} \alpha_{\min})}, \frac{1}{3 L_{\max} T} \right\}$. More precisely, after $R > 0$ communication rounds, the FedAvg Algorithm 1 satisfies*

$$\Phi(\underline{\boldsymbol{w}}^R) \leq \left( 1 - \frac{\eta T \alpha_g(\boldsymbol{w}^0, \rho)}{4} \right)^R \Phi(\underline{\boldsymbol{w}}^0). \tag{5}$$

---

**Essence of the Proof of Theorem 3.1:** Assumptions 2.1 and 2.4 lead to an exponential relation, specifically $\Phi(\underline{\boldsymbol{w}}^{r+1}) \leq \gamma^r \Phi(\underline{\boldsymbol{w}}^0)$, where $\gamma \in (0,1)$, (refer to Lemma F.4). To prove the existence of global optima $\boldsymbol{w}^*$ within the ball $\mathbb{B}[\underline{\boldsymbol{w}}^0, \rho]$, we have used the method of induction on two variables: global communication round $r$ and local updates $t$. By doing so, we conclude that the sequence $\{\boldsymbol{w}_k^{T,\tau}\}_{r,\tau \geq 0}$ remains confined within the ball $\mathbb{B}[\underline{\boldsymbol{w}}^0, \rho]$ (refer to Lemma F.6), which ensures that the sequence $\{\underline{\boldsymbol{w}}^r\}_{r=1}^{\infty}$ remains within the ball $\mathbb{B}[\underline{\boldsymbol{w}}^0, \rho]$ for all $r$. Further, we have shown that the sequence $\{\underline{\boldsymbol{w}}^r\}_{r=1}^{\infty}$ is Cauchy sequence in the closed subset $\mathbb{B}[\underline{\boldsymbol{w}}^0, \rho]$ of complete space. Therefore, it guarantees the limit of the sequence $\{\underline{\boldsymbol{w}}^r\}_{r=1}^{\infty}$, denoted by $\boldsymbol{w}^*$ belongs to the ball. A complete proof is provided in Appendix F. □

Note that Chatterjee (2022) required one condition to be satisfied for the linear convergence since their work considered a centralized setting. In contrast, our work requires two conditions for both global and local loss functions as stated in Assumptions 2.4 to guarantee linear convergence of FedAvg. Later we show that as the number of clients, $K$, increases, the requirement becomes more stringent. The above theorem leads to the following corollary.

**Corollary 3.2.** *By choosing $\eta$ as in Theorem 3.1, for any error $\epsilon > 0$, Algorithm 1 achieves a loss of $\Phi(\underline{\boldsymbol{w}}^R) < \epsilon$ after $R \geq \mathcal{O}\left( \left\lceil 2 \log\left( \frac{\Phi(\underline{\boldsymbol{w}}^0)}{\epsilon} \right) \right\rceil \right)$ communication rounds.*

Our next goal is to show that it is possible to initialize a NN such that it satisfies the conditions provided in Assumption 2.4. However, note that this does not provide any guarantees on the generalization error. To fill this gap, in the following sections, we consider a single hidden-layer NN and show that $(a)$ there exist an initialization and radius $\rho$ such that it results in a linear convergence leading to zero training loss (i.e., assumptions stated in Sec. 2 are satisfied), and $(b)$ prove that the generalization error can be made small by choosing large enough training samples and performing FedAvg for a sufficiently large number of communication rounds.

# 4 ASSUMPTION 2.4 FOR SINGLE HIDDEN LAYER NN WITH SQUARED ERROR LOSS

In this section, we show that there exist NNs such that Assumption 2.4 is satisfied, and hence leads to linear convergence of FedAvg (see Theorem 3.1). Towards this, we consider the following NN with a single hidden layer. In particular, we assume that the first layer has $m$ neurons followed by a smooth activation function. The output of this NN is given by Arora et al. (2019)

$$f_{\boldsymbol{w}}(\boldsymbol{x}) = \frac{1}{\sqrt{m}} \sum_{j=1}^{m} v_j \sigma(\boldsymbol{w}_j^\top \boldsymbol{x}), \tag{6}$$

where $\boldsymbol{x} \in \mathbb{R}^d$ is the input feature vector. With a slight abuse of notation, we have used $\boldsymbol{w} = \text{vec}([\boldsymbol{w}_1, \boldsymbol{w}_2, \dots, \boldsymbol{w}_m]) \in \mathbb{R}^{dm \times 1}$ to denote the aggregated weight vectors in the first layer and $\boldsymbol{v} = (v_1, v_2, \dots, v_m)^\top$ to denote the weight in the second layer, where $v_j \overset{\text{i.i.d.}}{\sim} \{-1, 1\}$. Now, we make the following assumption on the activation function.

**Assumption 4.1.** *We assume that $\sigma : \mathbb{R} \to \mathbb{R}$ is a smooth non-decreasing activation function such that $\sigma(0) = 0$. Further, first and second order derivatives of $\sigma$ are bounded i.e., $|\sigma'(x)| \leq D_\sigma$ and $|\sigma''(x)| \leq \Delta_\sigma$.*

Note that the above condition is satisfied by the $\tanh$ activation function, i.e., $\sigma(x) = \tanh(x)$. The condition $\sigma(0) = 0$ is assumed for the sake of simplicity and ease of notation. It turns out that, with random initialization, this can be relaxed without changing the main result of the paper. With $\sigma(x) \neq 0$, many activation functions such as Softmax, $\tanh$ to name a few (see Xu et al. (2015)) satisfy the conditions mentioned in Assumption 4.1. It is worth noting that the well-known ReLU activation does not satisfy the smoothness condition, but it can be well approximated by a smooth proxy function (see (Xu et al., 2015)).

**Assumption 4.2.** *Each node $k \in [K]$ samples $n$ i.i.d. data points denoted $\mathcal{X}_k = \{(\boldsymbol{x}_{k,1}, y_{k,1}), \dots, (\boldsymbol{x}_{k,n}, y_{k,n})\}$ from a continuous and possibly different distributions $p_k(\boldsymbol{x}), k \in [K]$ with $y_{k,i} \leq y_{max}$ for all $i \in [n]$.*

We consider the average loss function $\Phi(\boldsymbol{w}) := \frac{1}{K} \sum_{k=1}^{K} \Phi_k(\boldsymbol{w})$, where $\Phi_k : \mathbb{R}^{md} \to \mathbb{R}$ is the squared loss function for each client $k \in [K]$ and is defined as $\Phi_k(\boldsymbol{w}) = \sum_{i=1}^{n} [f_{\boldsymbol{w}}(\boldsymbol{x}_{k,i}) - y_{k,i}]^2 = \|\boldsymbol{e}_k\|_2^2$, where the $i^{\text{th}}$ entry of the error vector $\boldsymbol{e}_k := [f_{\boldsymbol{w}}(\boldsymbol{x}_{k,i}) - y_{k,i}]$. Using $\boldsymbol{e} = [\boldsymbol{e}_1, \boldsymbol{e}_2, \dots, \boldsymbol{e}_n]$, the global loss can be written as $\Phi(\boldsymbol{w}) := \frac{1}{K} \|\boldsymbol{e}\|^2$. Next, we discuss the conditions under which a single hidden layer neural network satisfies Assumption 2.4. It turns out that these conditions are dependent on the following Jacobian matrix:

$$\boldsymbol{J}_k(\boldsymbol{w}) = \frac{1}{\sqrt{m}} \times \boldsymbol{H}_k(\boldsymbol{w}), \tag{7}$$

where each entry of $\boldsymbol{J}_k(\boldsymbol{w})$ is a $d$-dimensional row vector, and $\boldsymbol{H}_k(\boldsymbol{w})$ is defined as follows

$$\boldsymbol{H}_k(\boldsymbol{w}) := \begin{bmatrix} v_1\sigma'(\boldsymbol{w}_1^\top \boldsymbol{x}_{k,1})\boldsymbol{x}_{k,1}^\top & v_2\sigma'(\boldsymbol{w}_2^\top \boldsymbol{x}_{k,1})\boldsymbol{x}_{k,1}^\top & \dots & v_m\sigma'(\boldsymbol{w}_m^\top \boldsymbol{x}_{k,1})\boldsymbol{x}_{k,1}^\top \\ \cdot & \cdot & \cdot & \cdot \\ \cdot & \cdot & \cdot & \cdot \\ \cdot & \cdot & \cdot & \cdot \\ v_1\sigma'(\boldsymbol{w}_1^\top \boldsymbol{x}_{k,n})\boldsymbol{x}_{k,n}^\top & v_2\sigma'(\boldsymbol{w}_2^\top \boldsymbol{x}_{k,n})\boldsymbol{x}_{k,n}^\top & \dots & v_m\sigma'(\boldsymbol{w}_m^\top \boldsymbol{x}_{k,n})\boldsymbol{x}_{k,n}^\top \end{bmatrix}, \tag{8}$$

where $k \in [K]$ and the size of the matrix $\boldsymbol{H}_k(\boldsymbol{w})$ is $n \times md$, i.e., $\boldsymbol{H}_k(\boldsymbol{w}) \in \mathbb{R}^{n \times md}$. We define a global Jacobian matrix $\boldsymbol{J}(\boldsymbol{w})$ by stacking $\boldsymbol{H}_k^\top(\boldsymbol{w})$ row-wise as $\boldsymbol{J}(\boldsymbol{w}) = \frac{1}{\sqrt{m}} \times [\boldsymbol{H}_1^\top(\boldsymbol{w}), \boldsymbol{H}_2^\top(\boldsymbol{w}), \dots, \boldsymbol{H}_K^\top(\boldsymbol{w})] \in \mathbb{R}^{md \times Kn}$. The following lemma provides a condition under which $\boldsymbol{J}_k(\underline{\boldsymbol{w}}^0)$ and $\boldsymbol{J}(\underline{\boldsymbol{w}}^0)^\top$ are full rank matrices. Note that the full rank requirement is only at the initialization. The size of the NN scales as $n/d$ as opposed to $n$ in (Chatterjee, 2022). This result is similar to the results of Zhang et al. (2021) but for an FL setting.

---
**Algorithm 2** FedAvg Algorithm for single hidden layer NN

---
1: **Initialization**: Initialize using $\underline{\boldsymbol{w}}^0 \sim \mathcal{N}(\boldsymbol{0}, \frac{1}{d}I_{md \times md})$ and $v_i \overset{\text{i.i.d.}}{\sim} \{-1, 1\} \ \forall i \in [m]$.
2: Broadcast $\underline{\boldsymbol{w}}^r$ to all the clients $k \in [K]$
3: Run the FedAvg Algorithm 1

---

**Lemma 4.3.** *At the random initialization $\underline{\boldsymbol{w}}^0 \sim \mathcal{N}(\boldsymbol{0}, \frac{1}{d}I_{md \times md})$, and $v_i \overset{\text{i.i.d.}}{\sim} \{-1, 1\}$ for all $i \in [m]$, the matrices $\boldsymbol{J}_k(\underline{\boldsymbol{w}}^0)$ and $\boldsymbol{J}(\underline{\boldsymbol{w}}^0)^\top$ have full column ranks almost surely provided $m \geq n/d$ and $m \geq nK/d$, respectively.*

*Proof:* The result follows by following the proof of Lemma $E.1$ of Zhang et al. (2021) for the matrices $\boldsymbol{H}_k(\underline{\boldsymbol{w}}^0)$ and $\boldsymbol{H}(\underline{\boldsymbol{w}}^0)^\top$. One main difference is that Zhang et al. (2021) uses mirrored Le-cun. However, the proof does not change for our initialization. □

Towards stating the condition for neural network, we need the following definitions

$$\lambda_{k,\rho}^-(m) := \inf_{\boldsymbol{w} \in \mathbb{B}[\underline{\boldsymbol{w}}^0, \rho]} \frac{\boldsymbol{e}_k^\top \boldsymbol{H}_k(\underline{\boldsymbol{w}}^0) \boldsymbol{H}_k(\underline{\boldsymbol{w}}^0)^\top \boldsymbol{e}_k}{\|\boldsymbol{e}_k\|^2}, \tag{9}$$

where $\boldsymbol{e}_k$ and $\boldsymbol{H}_k(\boldsymbol{w})$ are as defined earlier.[3] The following is an extension of the above definition to $K$ clients

$$\lambda_\rho^-(m) := \inf_{\boldsymbol{w} \in \mathbb{B}[\underline{\boldsymbol{w}}^0, \rho]} \frac{\boldsymbol{e}^\top \boldsymbol{H}(\underline{\boldsymbol{w}}^0)^\top \boldsymbol{H}(\underline{\boldsymbol{w}}^0) \boldsymbol{e}}{\|\boldsymbol{e}\|^2} \tag{10}$$

where $\boldsymbol{e} = [\boldsymbol{e}_1, \boldsymbol{e}_2, \ldots, \boldsymbol{e}_k]^\top \in \mathbb{R}^{nK}$ and $\boldsymbol{H}(\underline{\boldsymbol{w}}^0)$ is defined earlier. Similarly, $\tilde{\lambda}_{k,\rho}^-(m)$ and $\tilde{\lambda}_\rho^-(m)$ are defined by replacing $\boldsymbol{H}_k(\underline{\boldsymbol{w}}^0)$ by $\boldsymbol{H}_k(\boldsymbol{w})$ and $\boldsymbol{H}(\underline{\boldsymbol{w}}^0)$ by $\boldsymbol{H}(\boldsymbol{w})$ in equations 9 and equation 10, respectively. In addition, $\lambda_{\max}(\rho) := \sup_{\boldsymbol{w} \in \mathbb{B}(\underline{\boldsymbol{w}}_0, \rho)} \lambda_{\max}\left(\boldsymbol{H}(\boldsymbol{w})\boldsymbol{H}(\boldsymbol{w})^\top\right)$. These notations will be used in Theorem 4.5. Since we know from the above Lemma that the matrices $\boldsymbol{H}(\underline{\boldsymbol{w}}^0)\boldsymbol{H}(\underline{\boldsymbol{w}}^0)^\top$ and $\boldsymbol{H}_k(\underline{\boldsymbol{w}}^0)^\top \boldsymbol{H}_k(\underline{\boldsymbol{w}}^0)$, $k \in [K]$ are full rank, we next ask if the above terms scale with $m$. Recall that we are looking at the Jacobian to state the condition under which Assumption 2.4 is satisfied. Thus, the following assumption is important, whose analytical justification is provided in App. G.

**Assumption 4.4.** *We assume that both $\lambda_{k,\rho}^-(m)$ and $\lambda_\rho^-(m)$ scale linearly with $m$.*

**Experimental Justification of Assumption 4.4:** An observation similar to the above assumption was also made in (Telgarsky, 2021, page 39). We verify the above assumption via experiments in Fig. 1, where we have plotted the minimum eigenvalue of the Jacobian versus $m$ for different numbers of clients $K$ using the MNIST data set (LeCun & Cortes, 2010). We can observe from the figure that the variation is almost linear, and the slope increases with decreasing $K$.

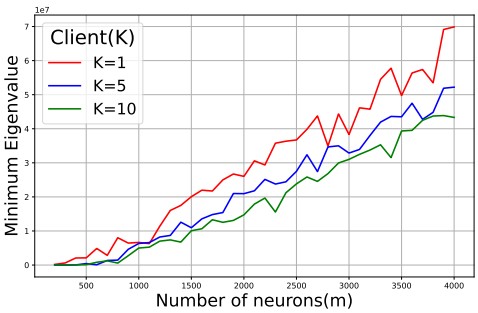

## 4.1 CONDITION ON NEURAL NETWORK (NN)

To prove the linear convergence of Algorithm 1 for single hidden layer NN, we need the definitions stated in equations 10 and 9. The following theorem provides a condition under which the Algorithm 1 converges linearly to a global optimal point, and the proof can be found in the Appendix I.

Figure 1: Plot of $\lambda_{\min}(m)$ versus $m$ for $K = 1, 5, 10$. Here, $K = 1$ corresponds to $\lambda_{1,\min}(m)$. This shows that Assumption 4.4 is valid in the real-world setting as well, i.e., the minimum eigenvalue scales linearly with $m$.

---
[3]Here, $\boldsymbol{e}_k$ and $\boldsymbol{e}$ depend on $\boldsymbol{w}$.

**Theorem 4.5.** *Let* $\Psi_{m,K,n,\rho} := \sqrt{bn\left(\frac{\lambda_\rho^+(m)}{m} + \frac{d\Delta_\sigma^2\rho^2}{m}\right)}$ *and* $b := \frac{2D_\sigma^2\rho^2 d\log(2n/\delta)}{m} + 2y_{\max}^2$,

*where* $\lambda_\rho^+(m) := \sup_{\boldsymbol{w}\in\mathbb{B}[\underline{\boldsymbol{w}}^0,\rho]} \frac{\|\boldsymbol{H}(\underline{\boldsymbol{w}}^0)\boldsymbol{e}\|^2}{\|\boldsymbol{e}\|^2}$. *The loss functions for single hidden layer NN satisfy equation 3 and equation 4 of Assumption 2.4 with a probability of at least* $1 - \delta/2$, *for any* $\delta > 0$ *provided the following holds:*

$$\frac{\lambda_{k,\rho}^-(m)}{m} > 2\times\left[\frac{\Delta_\sigma^2 d\rho^2}{m} + \frac{8bn}{\rho^2}\right], \text{ and } \frac{\lambda_\rho^-(m)}{m} > \frac{8K\Psi_{m,K,n,\rho}}{(1-\zeta_\rho)\rho} + \frac{2d\Delta_\sigma^2\rho}{m}, \quad (11)$$

*where* $\lambda_{k,\rho}^-(m)$ *and* $\lambda_\rho^-(m)$ *are as defined in equation 9 and equation 10, respectively.*

To the best of our knowledge, these conditions are the first of their kind. First, note that the terms $\lambda_{k,\rho}^-(m)/m$ and $\lambda_\rho^-(m)/m$ are less sensitive to $\rho$ since they are sandwiched between the smallest and the largest eigenvalues of $\boldsymbol{H}(\underline{\boldsymbol{w}}^0)^\top\boldsymbol{H}(\underline{\boldsymbol{w}}^0)$ and $\boldsymbol{H}_k(\underline{\boldsymbol{w}}^0)^\top\boldsymbol{H}_k(\underline{\boldsymbol{w}}^0)$, respectively. In particular, these eigenvalues depend on the initialization $\underline{\boldsymbol{w}}^0$ while the original condition is in terms of the ball around the initialization. Hence, using the eigenvalues in place of $\lambda_{k,\rho}^-(m)$ and $\lambda_\rho^-(m)$ in the new conditions makes it easy to verify (see Fig. 1). Secondly, the larger values of $\rho$ make the right-hand sides in the equation 11 large, and hence the conditions may not be satisfied, as expected. On the other hand, the same can be observed for smaller values of $\rho$ as well. Thus, a critical $\rho$ is necessary. By choosing $\rho = c \times \mathcal{O}(\sqrt{n})$ and $m = \mathcal{O}(n^3)$ in Theorem 4.5 ensures that the right hand sides scale down with $c$. Thus, the right-hand side is small for a large enough $c$. However, by Assumption 4.4, the left-hand sides, i.e., $\lambda_{k,\rho}^-(m)/m$ and $\lambda_\rho^-(m)/m$ are constants that depend only on the initialization (not on $\rho$), and do not scale with $m$ or $n$ or $c$. Hence, the conditions are satisfied for large enough $c$:

**Corollary 4.6.** *Choosing* $\rho = c\times\mathcal{O}(\sqrt{n})$ *and* $m = \mathcal{O}(n^3)$ *in Theorem 4.5 ensure that the conditions in equation 11 are satisfied for sufficiently large c.*

The above corollary shows that by choosing a large radius of $\rho$ and a large number of nodes in the second layer, linear convergence can be guaranteed. This brings in several challenges while proving the generalization guarantee, especially while proving a bound on the Rademacher complexity.

## 5 GENERALIZATION PERFORMANCE: SINGLE HIDDEN LAYER NN

In this section, we show that single hidden layer NN architectures exhibit impressive generalization guarantees. To state the generalization result, we need the following notion of Rademacher complexity of the single hidden layer NN.

**Definition 5.1 (See Mohri et al. (2019)).** *The Rademacher complexity of a class of single hidden layer NN constrained to a ball of radius* $\rho$ *at client* $k \in [K]$ *is defined as*

$$Rad_k(\underline{\boldsymbol{w}}^0, \rho) := \mathbb{E}_{|\boldsymbol{v}\in\mathcal{G}_{\boldsymbol{v}}}\left[\sup_{\boldsymbol{w}\in\mathbb{B}[\underline{\boldsymbol{w}}^0,\rho]} \frac{1}{n}\sum_{i=1}^{n}\zeta_i f_{\boldsymbol{w};\boldsymbol{v}}(\boldsymbol{x}_{k,i})\right],$$

*where the expectation is with respect to* $\boldsymbol{\zeta} := (\zeta_1, \zeta_2, \ldots, \zeta_n) \overset{i.i.d.}{\sim} \{-1, +1\}^n$, *conditioned on* $\boldsymbol{v} := (v_1, v_2, \ldots, v_m) \in \mathcal{G}_{\boldsymbol{v}} := \{\boldsymbol{v}\in\{-1,1\}^m : |\sum_{i=1}^{n}\zeta_i f_{\boldsymbol{w};\boldsymbol{v}}(\boldsymbol{x})| < \Delta\}$. *Here,* $\Delta := \sqrt{2}D_\sigma d\sqrt{\frac{\rho^2+m}{m}}\log 4$ *and* $\boldsymbol{x}$ *is any data point sampled from* $p_k(\boldsymbol{x})$.

For a FL setting, the generalization guarantee is provided in Mohri et al. (2019), and the result requires the loss to be bounded. However, in our case, the loss can potentially be unbounded. We handle this by focusing on the class of "good" NNs, i.e., $\boldsymbol{v} \in \mathcal{G}_{\boldsymbol{v}}$, whose output is bounded. In Appendix H, using the fact that the weight vector lies within a ball of radius $\rho$ around $\underline{\boldsymbol{w}}^0$, we show that there exists such NNs with bounded output. Subsequently, we show that for such NNs, the generalization is guaranteed. We use this result along with the result of Mohri et al. (2019) to show the following Theorem whose proof can be found in Appendix J.

**Theorem 5.2.** *Let* $\Psi := \left( (\rho^2 + 3m) \frac{2D_\sigma^2 d^2 \log 4}{m} + y_{max}^2 \right) \sqrt{2 \log(\frac{1}{\delta})}$. *For the single hidden layer NN with the initialization as in Algorithm 2 satisfying Assumptions 4.4 with $m \geq nK/d$, and the conditions of Theorem 4.5, with a probability of at least $1 - \delta$, the following inequality holds*

$$\Phi(\boldsymbol{w}; \boldsymbol{v}) \leq \Phi_S(\boldsymbol{w}; \boldsymbol{v}) + \frac{2n}{K} \sum_{k=1}^{K} Rad_k(\underline{\boldsymbol{w}}^0, \rho) + \Psi \sqrt{\frac{n}{K}}. \tag{12}$$

Recall that the loss function is defined as the sum of the loss on individual training samples. Thus, defining $\mathcal{L}(\boldsymbol{w}; \boldsymbol{v}) := \frac{\Phi(\boldsymbol{w}, \boldsymbol{v})}{n}$ and $\mathcal{L}_S(\boldsymbol{w}; \boldsymbol{v}) := \frac{\Phi_S(\boldsymbol{w}; \boldsymbol{v})}{n}$, and using this in the above theorem leads to the following.

**Corollary 5.3.** *For the single hidden layer NN with initialization as in Algorithm 2, with probability at least $1 - 2\delta$ over the draw of the samples $X_k \sim \mathcal{D}_k^n$, the following inequality holds*

$$\mathcal{L}(\boldsymbol{w}; \boldsymbol{v}) \leq \mathcal{L}_S(\boldsymbol{w}; \boldsymbol{v}) + \frac{2}{K} \sum_{k=1}^{K} Rad_k(\underline{\boldsymbol{w}}^0, \rho) + \frac{\Psi}{\sqrt{nK}}. \tag{13}$$

Next, we provide an upper bound on the Rademacher complexity.

**Theorem 5.4.** *The Rademacher complexity of client $k \in [K]$ is bounded by*

$$Rad_k(\underline{\boldsymbol{w}}^0, \rho) \leq \frac{1}{n\sqrt{m}} + \sqrt{\frac{\nu D_\sigma^2 d^2 (\log 4) \log(N_{\theta,\rho}/\delta_1)}{n}},$$

*where $\nu = (\rho^2 + 3m)/m$, $N_{\theta,\rho} := 3d^{3/4}\sqrt{\rho D_\sigma nm}$ and $\delta_1 := \frac{1}{2mn\sqrt{2}D_\sigma d} \sqrt{\frac{m}{\log 4(\rho^2+m)}}$.*

*Proof:* See Appendix K. $\qquad\qquad\square$

### 5.1 DISCUSSION

To the best of our knowledge, the above is the first result of its kind for an FL setup. We make the following remarks.

➢ The generalization error can be made small provided the right-hand side in the Corollary 5.3 is small. The first term, i.e., the empirical loss, depends on the communication rounds and the conditions stated in Theorem 4.5. The latter can be ensured by choosing $\rho = \mathcal{O}(\sqrt{n})$ and $m = \mathcal{O}(n^3)$, as shown in Corollary 4.6. In other words, the radius and the size of the NN scale with $n$ which is not desired in general. However, we believe that this cannot be eliminated unless we make some structural assumptions about the data.

➢ Note that $\delta_1$ and $N_{\theta,\rho}$ scale with $n$ and $m$. However, it appears as a logarithmic term, and hence, the Rademacher complexity does not grow linearly with $n$. The above choices of $\rho$ and $m$ ensure that the Rademacher complexity in Theorem 5.4 goes down as $\mathcal{O}(1/\sqrt{n})$. Also, the choice of $\rho$ cannot scale faster than $\sqrt{m}$.

➢ The last term in the generalization result scales down with $n$ as $1/\sqrt{n}$. Based on these observations, it is clear that the generalization error can be made small by choosing large enough communication rounds $R$ and the number of training samples $n$.

➢ Here, we present our theoretical insights on the effect of $K$. From the generalization bound in equation 5.3, it is evident that the last term decreases with $K$ as $1/\sqrt{K}$. However, for larger values of $K$, the learning rate is impacted by $K$ through $\frac{\zeta_p \rho}{T\sqrt{\Psi_0}}$, which scales as $1/\sqrt{K}$ (see Theorem 3.1). From equation 5, the loss goes down as $\exp\{-\mathcal{O}(R/\sqrt{K})\}$ leading to slower convergence. Thus, the overall effect of increasing $K$ on the generalization is insignificant; this is also demonstrated in our experimental results as well as several existing works.

The above argument shows that the average loss can be made small by choosing sufficiently large $m$, $n$, and communication rounds, as shown next.[4]

**Corollary 5.5.** *With a probability of at least $1 - \delta$, there exists a single hidden layer NN employing the FedAvg algorithm with sufficiently large $m$, $n$, and $R$ that achieves a small generalization error. More specifically, the generalization error goes down as $\mathcal{O}(1/\sqrt{n})$.*

## 6 EXPERIMENTAL RESULTS

In this section, we verify our theoretical findings with experiments performed on an NVIDIA DGX V100 machine. We have used an MNIST image data set LeCun & Cortes (2010) distributed across 5 and 200 clients. We have used the single hidden layer network model with 1000 neurons in the hidden layer and `tanh` activation function. In both cases, we have maintained around 50 data points at each client, which is less than the dimension of input feature vectors, i.e., around 1200, which satisfies the condition $d \geq n$ and $m \geq nK/d$. We execute FedAvg for $R = 500$ communication rounds along with $T = 5$ round of local updates at each client with i.i.d. data.

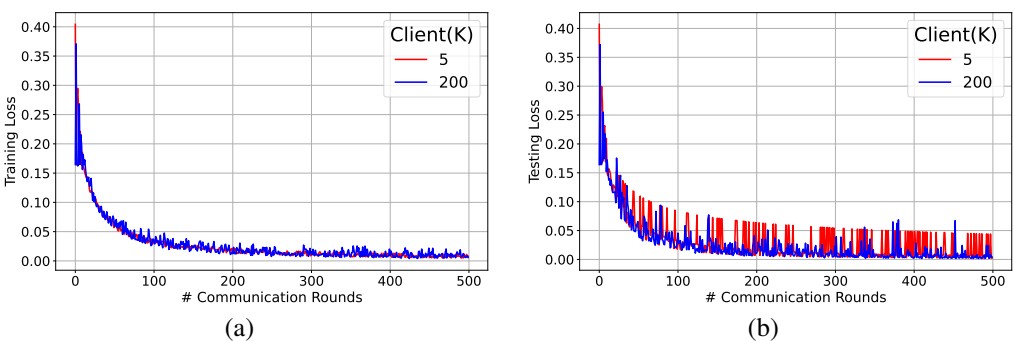

(a)  (b)

Figure 2: The Figures in (a) and (b) show the effect of the number of clients $K$ on the training and the testing losses, respectively. The experiments are done using MNIST data set.

Figure 2 shows the effect of $K$ on the testing and training errors. As suggested by our theory (see Sec. 5.1), increasing or decreasing $K$ has no effect on the performance (generalization and training loss).

## 7 CONCLUSIONS

In this work, we addressed the problem of generalization along with convergence guarantees of the widely used FedAvg algorithm for solving Federated Learning (FL) problems. We proved the generalization bound by handling the optimization error and the Rademacher complexity. The optimization error was handled by proposing a novel and new constrained Polyak-Łojasiewicz (PL) type conditions on the (local) loss functions. Under these new conditions, we showed that there exists a global optimum to which the FedAvg converges linearly after $\mathcal{O}(\log(1/\epsilon))$ rounds of communication, where $\epsilon$ is the desired optimality gap. Importantly, we demonstrated that a class of single hidden layer NNs satisfy the proposed conditions that are required to establish the linear convergence of FedAvg as long as $m > \frac{nK}{d}$, where $m$ is the number of neurons in the hidden layer, $n$ is the number of samples at each client, $K$ is the number of clients, and $d$ is the feature dimension. Finally, we showed that the generalization error of FedAvg decreases at the rate of $\mathcal{O}(1/\sqrt{n})$ by proving a bound on the Rademacher Complexity using the fact that the neural network parameters are constrained to a neighbourhood around the initialization.

---

[4]While stating this result, we have ignored $\log$ factors.

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
