# OpenReview forum: "Generalization of FedAvg Under Constrained Polyak-Lojasiewicz Type Conditions: A Single Hidden Layer Neural Network Analysis"
_ICLR.cc/2025/Conference — ICLR 2025 Conference Withdrawn Submission_

### Official Review · Reviewer_kk2S · 2024-10-20

**Soundness:** 2
**Presentation:** 3
**Contribution:** 2
**Rating:** 3
**Confidence:** 4

**Summary:**

The paper studies generalization and optimization of FedAvg with $K$ clients. To this aim, the paper first introduces a variant of PL condition called the constrained PL condition around the initialization point. Under some assumptions on loss function at each client and the global loss function, the paper first establishes a linear convergence rate for FedAvg. Then, the paper shows that these assumptions hold for a single hidden-layer neural network and therefore establishes a linear convergence for FedAvg with neural networks. Finally, the paper gives generalization bounds based on Rademacher complexity analysis. The generalization bounds are of order $O(1/\sqrt{n}+1/\sqrt{nK})$, where $n$ is the sample size.

**Strengths:**

- The paper introduces a local PL condition in a ball centered at the initialization point, and shows that under this local PL condition there exists a global minimum. This is better than assuming the existence of a global minimum in the literature.
- The paper shows that shallow neural networks with the least square loss satisfies this local PL condition. Furthermore, the generalization analysis does not require a bounded loss function, which is assumed in the literature.

**Weaknesses:**

- To establish the linear convergence, an important step is to show that the iterates stay within a neighborhood around the initialization point. However, it seems that the analysis towards this aim is problematic. For example, the proof of Theorem E.1 uses the inequality $\|w_k^{T}-w_k^0\|\leq \eta\sum_\tau\|\nabla \Phi_k\| $. Then, the paper uses the bound of $\|\nabla \Phi_k\|$ given in Lemma E.5. However, Lemma E.5 only holds in expectation. The underlying reason is that Lemma E.5 is based on Lemma E.4 and Lemma E.2. Note Lemma E.2 only gives bounds in expectation. Therefore, the first inequality in the proof of Lemma E.4 should involve an expectation over both sides. Therefore, the arguments in the paper only give bounds on $\mathbb{E}[\|w_k^{T}-w_k^0\|]$. This bound in expectation is not enough to apply the local PL condition to get linear convergence.
- The proof of Theorem 5.4 indicates that the covering number is of order $(3\rho\sqrt{d}/\theta)^d$. However, this covering number estimate seems to consider only a single $w_l$ instead of all nodes. The underlying reason is that the covering number estimate does not involve $m$.
- Below Eq (43), the paper uses the identity $\sigma(w_l^\top x)=\sigma(w_{l_j^\top x})+(w_{l_j}-w_l)^\top\nabla \sigma(w_{l_j^*}^\top x)$. This identity seems to not hold.
- As far as I see, the proof of Theorem 5.4 only gives Rademacher complexity bounds under the condition that $v\in \mathcal{G}_v$. It is not quite clear how to transfer this bound to Rademacher complexity bounds without this condition.

**Questions:**

- How to apply the bound of $\|w_k^{T}-w_k^0\|$ to get linear convergence?- Is the covering number estimate in the proof of Theorem 5.4 correct? Should it depend on $m$?- It would be nice if the authors give more details in the estimation of Rademacher complexities. For example, how does the condition on $v$ guarantee Rademacher complexity bounds?- The paper considers minibatch SGD in FedAvg. However, it seems that the analysis does not show the effect of $b$ in optimization and generalization. Then, the strategy of considering batch size larger than 1 is not justified.

**Details Of Ethics Concerns:**

Not applicable.

---

### Official Review · Reviewer_HAMs · 2024-10-28

**Soundness:** 3
**Presentation:** 2
**Contribution:** 2
**Rating:** 5
**Confidence:** 3

**Summary:**

The paper studies both the optimization and the generalization gap of the FedAvg under new Polyak-Łojasewicz type conditions. The authors derive results showing that, under certain assumptions, both the optimization and the generalization gap can be made arbitrarily small; that is, the generalization error (population risk) can be made small. In particular, the results are specialized to single hidden layer neural networks in order to derive more concrete conditions.

**Strengths:**

There are very few papers that simultaneously study the generalization gap and the optimization error in federated learning. The paper is particularly strong in this regard. The overall technical level is also good.

**Weaknesses:**

Unfortunately, there are several shortcomings in the paper. Please refer to questions for the detail of the below points.

- There are several inaccurate statements in the paper.

- The paper overlooks the literature on the generalization gap of federated learning algorithms.

- Some of the main claims of the paper are not sufficiently justified and contradict previous work. For example, the claim that the number of clients has no significant impact on the (especially) generalization gap of FedAvg.

- The derived generalization bounds do not reflect the behavior with respect to many previously reported factors, such as the number of clients, rounds, and the degree of heterogeneity. Moreover, the proof technique is not new and consists of classical $\epsilon$-net covering methods to bound the Rademacher complexities of each client; independent of the other clients and the FedAvg algorithm.

- The derived orders for $\rho$ and $m$ cause the NN output to change order-wise with $n$.

- The simulation part is disappointing.

**Questions:**

**Imprecise statements/missing justifications or inuitions:**

1.	In the abstract, it’s stated that ``both Rademacher complexity and the generalization error of FedAvg decrease at an optimal rate of $\mathcal{O}(1/\sqrt{n})$.’’ Why is it optimal? Where has it been shown?

2.	In the abstract it’s stated that “We further show that increasing the number of clients K decreases the generalization error at the rate of $\mathcal{O}(1/\sqrt{n}+1/\sqrt{nK})$.” This statement is not rigorous, since by definition $\mathcal{O}(1/\sqrt{n}+1/\sqrt{nK})=\mathcal{O}(1/\sqrt{n})$. Hence, no decrease of the generalization error w.r.t. number of clients is shown. However, at least the generalization gap is expected to decrease with the number of clients, as shown in some previous work (see below in the missing references).

3.	What are the “several existing works” that show increasing $K$ have insignificant impact on the generalization (as mentioned in line 485)?

4.	Can authors justify lines 68-70?

5.	More importantly, the authors should give some intuitions about the important assumption 2.4. Is it just a technical assumption? The authors claim that this is a relaxed condition w.r.t. previous ones; however, they do not provide any justification that such conditions are more relaxed. The fact that "our proposed condition does not require this assumption; instead, we prove the existence of a global optimal point under our novel condition" does not necessarily imply that the assumptions made are more relaxed.

6.	If there is a strong difference between Assumption 2.4. and the PL condition (as stated by the authors on page 4), then how would the fact that the PL inequality is shown to be satisfied for the wide neural networks be a convincing evidence that such networks satisfy Assumption 2.4 (sated at the top of page 5)?

7.	Can authors justify the claim in lines 391-392?

8.	Why choosing $1/2$ is without loss of “optimality” in line 1402?

9.	As mentioned by the authors, "the radius and size of the NN scale with n, which is generally undesirable". The authors then mention that "however, we believe that this cannot be eliminated unless we make some structural assumptions about the data". In the absence of any justification or reasoning, such subjective sentences make the paper imprecise.

**Missing references:**

10.	At least the following references are missing:

-	Information-theoretic bounds on the generalization error and privacy leakage in federated learning. ISIT 2020.

-	Improved information theoretic generalization bounds for distributed and federated learning. ISIT 2022.

-	Rate-distortion theoretic bounds on generalization error for distributed learning. NeurIPS 2022.

-	More Communication Does Not Result in Smaller Generalization Error in Federated Learning. ISIT 2023.

-	Why (and When) does Local SGD Generalize Better than SGD? ICLR 2023

-	Federated Learning with Nonvacuous Generalisation Bounds, 2023

-	Lessons from Generalization Error Analysis of Federated Learning: You May Communicate Less Often! ICML 2024

-	Can We Theoretically Quantify the Impacts of Local Updates on the Generalization Performance of Federated Learning? MobiHoc 2024

11.	Also, for the optimization analysis, comparison with recent results is required rather than comparison with older results such as (Haddadpour et al., 2019; Stich, 2018).


**Generalization Gap:**

12.	In the above references, there were some works showing that the "generalization gap", i.e. $\mathcal{L}(w;v)-\mathcal{L}\_S(w;v)$ in Corollary 5.3, should decrease as a function of $nK$ (or even sometimes as $nK^2$). This is indeed expected, since by increasing the number of clients, we increase the number of existing sample points. This behavior is not captured by the results of Section 5. In particular, these results ``suggest'' that if we fix $n$ and let $K\to \infty$, the generalization gap will not vanish; this seems to be incorrect both intuitively and by previous work.

13.	Another aspect; how does the proposed generalization gap behave with respect to the number of communication rounds? It is expected that such a gap increases with the number of communication rounds. (for a fixed total number of data samples).

14.	Moreover, the derived generalization gap seems to be independent of the heterogeneity of the data across clients (in contrast to e.g. Sun et al. (2023)). This is another important shortcoming of the paper's results.  This shortcoming also seems to apply to the optimization analysis.

15.	The generalization gap analysis seems to be independent of the previous section, except for the fact that the learned model is inside $\mathcal{B}[\underline{w}^0,\rho]$. The derived results are classical results that depends on the dimension of the model space. The $\mathcal{O}(1/\sqrt{n})$ could be established in particular by imposing the norm of the model to be bounded by $\rho \propto \sqrt{n}$.

**Simulations:**

16.	In general, I believe not all papers need extensive simulations; but here 1) the simulations are extremely simple, 2) no code is provided, 3) more importantly, the paper relies on a considerable number of assumptions and provides only upper bounds on the performance (I know the lower bounds are not easy to establish), so the simulations are necessary to validate different parts of the results; including the optimization results, the generalization gap results, and the overall results.



**Others:**

17.	What is in practice the order of the learning rate that satisfy the condition of Theorem 3.1.?

18.	If I got it correctly, the authors in Definition 5.1. and the results based on this definition, consider the cases where the output of the NN is bounded. Do they consider the probability of such event?

19.	Proof of Lemma H.1. As far as I understand, this lemma states that there exists a $\nu$ such that the output of the NN for **any** $x$ and $w$ (in the ball of radius $\rho$) is bounded by $\Delta$. However, what is shown is
$$Pr \left( \exists \nu \in \\{-1,1\\}^m : |f\_{w,v}(x)| \leq \Delta \right).$$
This would show that for any $x$ (and $w$?), there exists a $\nu$ with bounded $\Delta$. Can you clarify this?

20.	Based on the orders derived in Corollary 4.6, we should have $\\|\mathbf{w}_i\\| = \mathcal{O}\left(n^{-5/2}\right)$.  This does not seem to be a reasonable assumption. Similarly, the final output of the neural network seems to change order-wise, as $n$ grows, and I guess the loss function will increase. Can you clarify this order-wise choices?


**Minor comments:**

-	It is recommended to give some examples/justifications for Assumptions 2.1. and 2.2.

-	I found Q1 is not really a general motivating question; but rather something that the proof techniques of this paper rely on.

-	Note that Q2 has been addressed in some works, as mentioned above.

-	For better readability, the dependency of $\alpha(\underline{w}^0,\rho)$ on $f$, could be considered in the notation; e.g. $\alpha_f(\underline{w}^0,\rho)$. Especially, since $\alpha$ is considered for different functions.

-	Denoting the set of pairs $(x,y)$ by $\mathcal{X}$ does not seem to be a good notation (in Assumption 4.2)

-	On line 300; shouldn’t the loss functions $\Phi_k$ depend also on the training dataset $\mathcal{X}_k$?

-	Lines 301-303 are not clear enough. I imagine that the index $k$ refers to the client, but then I am not sure how $e=[e_1,\ldots,e_n]$ is defined. It seems that $e_k= [e_{k,1},\ldots,e_{k,i},\ldots,e_{k,n}]$ where $e_{k,i}=f_w(x_{k,i})-y_{k,i}$. Furthermore, $e_i = [e_{1,i},\ldots,e_{k,i},\ldots,e_{K,i}]$. Is it correct?

-	$\mathbf{J}_K$ is defined proportional to $\mathbf{H}_k(w)$, but $\mathbf{J}$ proportional to concatenation of $\mathbf{H}_k(w)^T$.  Better to be make them consistent.

-	In Lemma 4.3; doesn’t the condition $m\geq nK/d$ include the other condition $m\geq n/d$?

-	In line 420, I guess $\zeta_i$ have uniform distribution.

-	Typo line 349: “equations 9 and equation 10”

---

### Official Review · Reviewer_7ctU · 2024-11-03

**Soundness:** 3
**Presentation:** 2
**Contribution:** 3
**Rating:** 6
**Confidence:** 3

**Summary:**

This  paper investigates the generalization performance of Federated Averaging (FedAvg), a commonly used algorithm in Federated Learning (FL), which allows multiple clients to collaborate on a machine learning task without sharing their data. The paper introduces novel constrained Polyak-Łojasiewicz (PL)-type conditions that guarantee the linear convergence of FedAvg to a global optimal solution, thereby improving generalization performance. It then proves generalization bounds for single hidden layer neural networks, demonstrating that generalization error decreases with the number of training samples and neural network size. The study also explores the impact of the number of clients on generalization performance and convergence rate.

**Strengths:**

-  The authors tackle this challenge by carefully bounding both the optimization error and the Rademacher complexity  concurrently.
- The authors introduce new constrained PL-type conditions (Assumption 2.4) that differ from the typical assumptions of strong convexity or global PL conditions often used in optimization literature. These conditions are crucial for proving the linear convergence of FedAvg (Theorem 3.1) and play a significant role in bounding the optimization error.
- The authors provide a detailed analysis of single hidden layer neural networks, establishing exact conditions on model size (hidden layer neurons), per-client sample count, and Jacobian matrix eigenvalues for their theoretical results. This analysis highlights how these factors affect FedAvg’s convergence and generalization within this model class.

**Weaknesses:**

- The paper only specifically analyzes single hidden layer neural networks, and the findings may not directly apply to other model architectures.
- The analysis relies on certain assumptions, such as the scaling of eigenvalues of the Jacobian matrix, which have been experimentally verified but may not hold universally.
- The paper focuses on a theoretical analysis of these conditions and their experimental validation on the MNIST dataset. Further research is needed to explore their applicability to other model architectures and datasets.
- The conditions for achieving linear convergence and good generalization require the radius $\rho$ and the model size to scale with the number of samples. While the paper acknowledge this aspect, choosing a large radius and model size might have practical limitations in real-world applications
- The results need more elucidation.

**Questions:**

- It is not clearly explained in the main text why the exisiting works need the PL-condition.
- In the abstract, the last sentence shows that the number of clients actually has no effect on the convergence rate, which is also explained in Section 5. Hence, this sentence in the abstract is confusing. Futhremore, it would be beneficial to explain the advantage of increasing or decreasing the number of clients if it has not effect on the generalization error.
- It is not surprising to get that the Rademacher complexity is of $O(1/\sqrt{n})$. Could the authors compare it with other related works and explain why the result is important here?
- In Line 145, the sentence ends with "Ji & Telgarsky (2018)". Does it mean that the assumptions are adopted from their paper?
- Please elucidate the meaning of Assumption 2.4 first before comparing with other exising assumptions. Now it is hard to parse Assumption 2.4.
- The paper emphasizes the linear convergence but it is not clear what it is linear in.
- For my understanding, Theorem 3.1 provides an implicit bound on the optimization error. Could the authors elucidate more about this point?

---

### Official Review · Reviewer_P16N · 2024-11-06

**Soundness:** 2
**Presentation:** 3
**Contribution:** 2
**Rating:** 3
**Confidence:** 3

**Summary:**

The paper a new PL type condition over the local and global losses of the federated learning system that allows for considering a wider class of losses and which also implies the existence of a global minimum. Under the square loss, specific choices of radius around random initialization are set, as well as the width and further conditions that a single-layer neural network must have in order to satisfy such new PL type condition. Generalization results are also provided using a classic Rademacher complexity approach for square losses. Simulation to elucidate the effect of number of clients is also provided.

**Strengths:**

- The paper is well-motivated. It is a good contribution to think on new conditions that allow for training and generalization guarantees for a wider class of losses.
- The paper considers a neural network as function approximator for FedAvg, which is relevant in practice.
- The paper is well-organized in terms of the sequential presentation of its results.

**Weaknesses:**

I detail all the found weaknesses in the paper below. As a result, I believe the paper is not ready for publication yet.

Regarding the theoretical results:
- Regarding Assumption 2.1: It seems that the “$L_k$-smoothness imply the $L-smoothness$ with $L=\frac{1}{K}\sum^K_{k=1}L_k$. If this is the case, why is there a need to introduce two types of smoothness in the assumption when one implies the other?
- Regarding Assumption 2.2: Why is the smoothness definition in Assumption 2.2 so different than Assumption 2.1, i.e., without being expressed as the norm of a difference of gradients?
- Moreover, there is no mentioning of Assumption 2.1 in Theorem 3.1, nor in any of the other results’ statements. Where is it used?
- Also, it is my understanding that we are always dealing with *empirical losses* when we are trying to provide optimization convergence and not with *population losses* (population losses are only important for generalization results). Then, why do we really need Assumption 2.1 which is only in terms of population losses (e.g., see the definition of $\Phi_k$ which is in terms of *expectations over the true distribution of the data* instead of *an average of over the data samples*)? Why is Theorem 3.1, which should deal with data and empirical measures, considering Assumption 2.1 instead of Assumption 2.2?
- My other concern is with the existence of Assumption 4.4. This assumption is only “used” to explain a little remark about the conditions in equation (11). However, it is not an assumption that is explicitly stated in any of the theoretical results of the paper. I think perhaps the authors had in mind to consider Assumption 4.4 for Corollary 4.6, in which case, this should be made explicit. In fact, I would just remove Assumption 4.4 and place its content inside the Corollary 4.6’s statement. Then, the simulations of Figure 1 can be placed afterwards.
- Corollary 4.6. seems incorrect. According to my calculations, the choices of $\rho$ and $m$ would provide: $$\frac{\lambda}{m}>\mathcal{O}\left(\frac{c^2}{n^2}+\frac{1}{c^2}\right) $$.
If we assume Assumption 4.4, then the left-hand side is a constant. The corollary claims that the above inequality is satisfied for a sufficiently large $c$. However, a large $c$ will make the right-hand side grow unbounded, which could easily become (or be) larger than the constant at the left-hand side and thus violate the inequality. Unless I am missing something or the authors omitted something I am not considering, Corollary 4.6 is incorrect.
- In the third point of Section 5.1, it is claimed that certain scaling occurs “by choosing large enough communication rounds $R$”. However, I don’t see the explicit dependence on $R$ in equation (13), nor in Theorem 5.4. Why are the authors making such claim?


Important concerns regarding the contribution of the paper:
- In the Introduction, works such as Haddadpour et al. (2019), Stich, 2018, Qu et al., 2020, are mentioned to impose strong conditions on the losses because of the type of PL conditions or strong convexity being assumed. However, most of the results of the paper are based on the square loss. Indeed, the paper does not provide concrete examples of other losses used in practice for which Assumption 2.4 holds. Are there any other losses? Also, has the square loss been used in other previous works? This last question is very important because it delimits better the scope of the paper’s contribution, and there is currently no answer to these questions in the paper.
- The contributions and introduction should make explicit that the results using the one-layer neural network and generalization ones are **only** under the **square loss**. When I started reading the paper I thought that the classes of losses under which the main results hold were more general, but, as a reader, I was misled. The authors should make the contributions’ details more specific.
- In the paragraph that starts at line 255, the authors show how their work requires *two* conditions on local and global losses, as opposed to one condition as in the work by Chatterjee (2022). Can the authors provide intuition as to why two conditions are needed in their approach? Why isn’t just a condition on the local loss enough?
- The simulation is lacking important plots to strengthen the theoretical findings. For example, I suggest setting $m=c_1\cdot\frac{nK}{d}$ and $d=c_2\cdot n$ for some constants $c_1,c_2\geq1$, and then choose three or four values of $n$ (e.g., 20, 200, 2000) and plot how training and testing losses change across sample sizes and see if **all of them converge to the global minimum** (zero training loss), which is what I would expect from the theoretical results.

There are many notation problems that may have strong repercussions in the understanding of the paper and which show that paper needs to have further proofread.
- Around line 138 the author introduced a notation for the “empirical loss function (…) using sample $i$”, however it makes no sense to talk about an empirical loss function if it doesn’t include all the samples from the sample set. Besides, the authors do not provide a formula for $\Phi_{k,i}$ which they should in order to clarify what they mean by this expression.
- What is $\alpha_g$ in Assumption 2.4 and Theorem 3.1? This symbol has not been defined anywhere in the paper (the symbol $\alpha$ has been introduced, but not $\alpha_g$).
- In Remark 1. Shouldn’t it be $\Phi_{k,i}(\mathbf{w}^*)$ instead of $\Phi_{i}(\mathbf{w}^*)$?
- In Theorem 3.1’s statement and in the third point of the Section 5.1: Why is “$R>0$” being stated? Shouldn’t it be “$T>0$” according to Algorithm 1, i.e., the number of communication rounds? The symbol $R$ is not defined.
- In parts of the paper, such as after equation (6) and Lemma 4.3, the authors use the notation “$v_j\overset{i.i.d.}{\sim}\\{-1,+1\\}$”; however, this is not a correct notation since “$\\{-1,+1\\}$” is not a distribution, but a set. The authors must define that they want a distribution whose support is the set $\\{-1,+1\\}$. Another problem with this bad notation is that it is used in Definition 5.1 when defining the **Rademacher complexity**: the problem with this is that for such complexity we want a Rademacher distribution, not just *any* distribution over $\\{-1,+1\\}$.
- The symbol $\mathbf{H}$ after equation (10) has not been introduced before. It must be introduced.

Things to clarify:
- Assumption 2.4 deals with population losses, thus, its statement should be more explicit about giving the definition of $\Phi_k$ again, to have in mind that it is *dependent* on the underlying data distribution. In fact, the paper does not stress how these types of assumptions are *dependent on the data distribution*. As an alternative, I suggest including in the same equation (1) the definition of $\Phi_k$ because it is highly important: currently, the definition of $\Phi_k$ is buried in the text after equation (1).
- Following the previous point: do previous works also make conditions that are dependent on the data distribution?
- The fact that many results are under square loss is almost buried in the paragraph after Assumption 4.1. The authors should place the assumption of the square loss in either a new assumption environment or put it with Assumption 4.1. This is an important assumption, and the paper currently seems to just belittle it but not giving it the proper attention. This new assumption of square loss should be explicitly considered in the statements of all theoretical results that use it.
- What initialization is used for $v_i$ in the linear scaling experiments (the ones described in Figure 1)?


Other issues:
- Line 215: it should say “we will show” instead of “we have shown”.
- First paragraph of Section 3: it should say “Assumption 2.4” instead of “equation 2.4”.
- The term $\max_{k}l’_k$ can be removed since it doesn’t really add anything to the theorem’s statement.
- Please, double check where citations should be within parentheses and when it shouldn’t be.
- Sourav is a male name. Line 256 should say “his work” instead of “their work”.
- In the paragraph after Assumption 4.1, the *tanh* function is said to be $\sigma(0)\neq 0$: this is not true.
- Corollary 5.3 seems more important than Theorem 5.2 because it states a result comparing the losses, i.e., the generalization result in itself. Thus, I suggest “framing” this corollary instead of Theorem 5.2.
- $\Phi_S$ used in Theorem 5.2 needs to be defined.

**Questions:**

Please, see the Weaknesses question.

---

### Note · Authors · 2024-11-28

I have read and agree with the venue's withdrawal policy on behalf of myself and my co-authors.